# Interdigital and Plantar Foot Infections: A Retrospective Analysis of Molecularly Diagnosed Specimens in the United States and a Literature Review

**DOI:** 10.3390/microorganisms13010184

**Published:** 2025-01-16

**Authors:** Aditya K. Gupta, Tong Wang, Sara A. Lincoln, Wayne L. Bakotic

**Affiliations:** 1Division of Dermatology, Department of Medicine, Temerty Faculty of Medicine, University of Toronto, Toronto, ON M5S 3H2, Canada; 2Mediprobe Research Inc., London, ON N5X 2P1, Canada; twang@mediproberesearch.com; 3Bako Diagnostics, Alpharetta, GA 30005, USA; slincoln@bakodx.com (S.A.L.); wbakotic@bakodx.com (W.L.B.)

**Keywords:** erythrasma, intertrigo, tinea pedis, interdigital tinea pedis

## Abstract

Up to one-quarter of the United States population is affected by tinea pedis (athlete’s foot). Tinea pedis of the web space (interdigital tinea pedis) is a common clinical presentation causing skin macerations and fissures. A “dermatophytosis complex” (i.e., concomitant bacterial colonization) further complicates treatment. Here, we examined records of 14,429 skin specimens taken from the feet of dermatology and podiatry outpatients over a 4.6-year period; all specimens were subjected to multiplex qPCR diagnosis for the detection of dermatophytes, *Candida*, *Corynebacterium minutissimum*, *Pseudomonas* and *Staphylococcus aureus*. A literature search was conducted to review the reported prevalence of fungal and bacterial agents. In both interdigital and plantar foot specimens, dermatophytes (33.3–33.8%) and *S. aureus* (24.3–25%) were found to be the predominate pathogens. In the interdigital space, a higher prevalence of *C. minutissimum* (15.7% vs. 7.9%) and *Pseudomonas* (23.5% vs. 9.6%) was found. The detection of *Pseudomonas* was more likely to be observed in the presence of *Candida*, reflecting a higher risk of mixed infection. In dermatophyte-positive specimens, the “dermatophytosis complex” variant was observed at 45.5% (SD: 2.3). An analysis of patient characteristics showed male patients exhibiting higher likelihoods for dermatophyte, *C. minutissimum*, *Pseudomonas* and *S. aureus* detections. The elderly were disproportionately infected with *Candida*. In children, an *S. aureus* detection was more common, which could be attributed to impetigo. The recent literature lacks reporting on concomitant bacterial colonization in tinea pedis patients, likely due to the reliance on fungal culture supplemented with antibiotics. Geographical variation has been identified in the detection of the *Trichophyton mentagrophytes* complex. In conclusion, PCR diagnosis serves as a valuable tool for the management of tinea pedis. An accurate and timely detection of fungal pathogens and concomitant bacterial colonization can better inform healthcare providers of appropriate treatment selection.

## 1. Introduction

There has been a significant increase in the incidence and prevalence of superficial fungal infections over the past three decades [1]. Globally, an estimated 7790 per 100,000 persons were affected in 2021 [1]. Tinea pedis (athlete’s foot) has emerged in part due to the growing trend in urbanization, comorbidities’ burdens and an aging population, as well as sports and other recreational activities involving occlusive footwear [2]. In the United States, tinea pedis may be found in up to one-quarter of the population [2], with disproportionate impacts on ethnic minorities, low-income individuals, the elderly, and those living with physical disabilities [3].

Clinically, tinea pedis typically presents as skin maceration of the web spaces (i.e., interdigital tinea pedis) located between the third and fourth toes, or the fourth and fifth toes, which can lead to erythema and skin fissures [2,4]; adjacent skin surfaces (e.g., sulcus) may also be affected. Moccasin tinea pedis, characterized by dry, scaly lesions and fissures on the plantar side of the foot, is the second most common clinical presentation [2]. Vesicular tinea pedis and acuate ulcerative tinea pedis are less common. The former is characterized by painful, pruritic and erythematous lesions with small vesicles on the instep or the medial plantar surface, and the latter is characterized by ulcers and erosions of the toe web spaces due to the exacerbation of interdigital tinea pedis [2]. If left untreated or undertreated, these infections can spread to adjacent nails, causing onychomycosis [2]. Chronic lesions, as a result of the breakdown of the skin barrier or the nail unit, can increase the risk of secondary bacterial colorizations/infections, which is especially concerning for diabetic patients with neuropathy [5].

Differential diagnoses for tinea pedis include non-infectious etiologies (e.g., psoriasis, atopic dermatitis or shoe contact dermatitis), intertrigo due to bacterial or *Candida* infections, and erythrasma due to *Corynebacterium minutissimum* [2]. Prior to initiating antifungal treatments, it is recommended to conduct confirmatory testing for dermatophytes (*Trichophyton*, *Epidermophyton*, *Microsporum*), which includes traditional techniques such as direct microscopy and fungal culture and newer techniques such as polymerase chain reaction (PCR) with an improved sensitivity [2,3]. In cases of tinea pedis complicated by concomitant bacterial colonization/infection (i.e., “dermatophytosis complex”) [4,5], patients may present with severe symptoms, including malodor, pruritus, inflammation and skin maceration, which may warrant the use of antifungal and antibacterial agents. However, conducting fungal culture with the addition of antibiotics (e.g., chloramphenicol)—to compensate for the slow growth rate of dermatophytes—means that cases of “dermatophytosis complex” are likely under-reported, as opposed to PCR diagnosis, which does not require the growth of an isolate. In the present study, we reviewed multiplex qPCR diagnostic records of interdigital and plantar foot specimens obtained from patients in the United States from 2020 to 2024, with the aim of updating our current understanding of the pathogen distribution and patient risk factors. A literature search was conducted to review the global prevalence of fungal and bacterial agents in suspected tinea pedis patients.

## 2. Materials and Methods

Diagnostic records of 14,429 specimens were reviewed, consisting of skin scrapings obtained from patients with suspected infectious dermatitis localized to the plantar side of the foot (sole, arch, heel, instep) or the toe web space (web space, sulcus). All specimens were submitted by dermatologists and podiatrists across the United States to a CLIA-certified laboratory (Bako Diagnostics, Alpharetta, GA, USA). The present study represents a retrospective review of secondary data which were deidentified. PCR diagnostic testing was provided based on the treating physician’s order as part a non-interventional, standard-of-care procedure. Thus, this study does not constitute a clinical trial for which ethics overview and approval would be required.

Skin scrapings were collected at the point-of-care using a sealed specimen envelope (Dermapak 2000, DERMACO Ltd., Bedford, UK) or other sterile container with a tightly fitted cap, followed by transportation at ambient temperatures. All specimens were initially subjected to DNA extraction, which includes the following steps: (1) lysis in a beaded tube, (2) homogenization (Bead Mill Homogenizer, Omni International, Kennesaw, GA, USA), (3) incubation and centrifugation, and (4) DNA extraction (Mag-Bind Plant DNA DS kit, Omega Biotek, Norcross, GA, USA) using a Hamilton Microlab STAR workstation (Reno, NV, USA). DNA elutes were then subjected to a multiplex qPCR assay for the detection of pan-dermatophytes, *Candida*, *C. minutissimum*, *Pseudomonas* and *S. aureus*.

Data were tabulated using Microsoft Excel (version 2301). Recorded data parameters include patient demographic variables (i.e., age, sex), clinic location by state, sampling site and submission date, and multiplex qPCR results. Toe web space specimens were identified using the following sampling sites: interspace, digit and sulcus; foot specimens were identified using the following sampling sites: foot, plantar, arch, heel, lower extremity and instep. Location data were classified into one of four US Census Regions: Northeast, Midwest, South and West. Pathogen identification results were summarized using the mean and the standard deviation (SD). Analyses of pathogen-identified results, between-group and within-group, were performed using one-way ANOVA and paired sample *t*-tests (IBM SPSS Statistics 20). Potential patient risk factors were identified using the odds ratios (ORs) with 95% confidence intervals (CIs); 2-sided *p*-values were calculated as previously described by Altman and Bland [6]. A *p*-value of <0.05 was considered statistically significant.

### Systematic Literature Search

A literature search was conducted on October 25, 2024, on PubMed, Embase (OVID) and the Cochrane Library. The search terms and subject headings included the following: ‘interdigital’, ‘toe web space’, ‘tinea pedis’, ‘athlete foot’, ‘erythrasma’, ‘intertrigo’, ‘dermatophyte’, ‘*Candida*’, ‘*Corynebacterium minutissimum*’, ‘*Pseudomonas*’, ‘*Staphylococcus aureus*’. Our inclusion criteria were clinical investigations (2010–2024) that identified fungal or bacterial agents causing interdigital or plantar foot infections, using culture or molecular methodologies. Information regarding pathogen identification and prevalence was extracted for review. Studies that exclusively examined the inpatient population or a comorbid population (e.g., diabetic foot ulcers, HIV) were excluded; outpatient studies that compared one comorbid patient group against a control group were included. Case reports, non-English articles and articles that did not identify the etiological agent or that are without complete reporting of the identification results were excluded; studies that solely relied on direct microscopic examinations for pathogen identification were excluded. Reviews, expert opinions and conference proceedings were also excluded.

## 3. Results

### 3.1. Microorganisms

Analysis included 10,473 interdigital (web space) foot specimens and 3956 plantar foot specimens submitted from January 2020 to August 2024 (4 years and 7 months). The most commonly detected pathogens in interdigital and plantar foot infections were dermatophytes (interdigital: 33.3% [SD: 3.2]; plantar: 33.8% [SD: 3.6]) and *S. aureus* (interdigital: 25.0% [SD: 2.1]); plantar: 24.3% [SD: 2.2]) (Figure 1). Compared to plantar foot infections, *C. minutissimum* (15.7% [SD: 3.9] vs. 7.9% [SD: 2.7], *p* < 0.01) and *Pseudomonas* (23.5% [SD: 5.2] vs. 9.6 [SD: 5.4], *p* < 0.01) were significantly more likely to be detected in interdigital foot infections. No significant differences in detection rates were observed for dermatophytes, *Candida* and *S. aureus*.

### 3.2. Patients’ Influence

Dermatophytes were more likely to be observed in males than females (Table 1); for interdigital and plantar foot specimens, males exhibited a 10% (OR: 1.1 [95% CI: 1.1, 1.2]; *p* = 0.001) and a 40% higher likelihood (OR: 1.4 [95% CI: 1.2, 1.6]; *p* < 0.001) than females, respectively. For *Candida* detections, no significant differences between the sexes were observed. With regard to age, a dermatophyte detection, both interdigital and plantar, was most likely to be observed in young adults between the age of 18 and 44 (interdigital specimen: 40.5%; plantar specimen: 38.9%). Compared to young adults, plantar foot specimens from children were observed to have a 70% lesser likelihood of dermatophyte detections (OR: 0.3 [95% CI: 0.2, 0.5]; *p* < 0.001). In contrast, *Candida* detections were more frequently observed in older age groups. Compared to the 18–44-year age group, interdigital specimens from children were 90% less likely to show *Candida* (OR: 0.1 [95% CI: 0.04, 0.6]; *p* = 0.007), whereas an 80% higher likelihood (OR: 1.8 [95% CI: 1.5, 2.2]; *p* < 0.001) was observed in the interdigital specimens from the elderly (≥65 years). A similar trend was observed in plantar foot specimens, where *Candida* detections were 80% (OR: 1.8 [95% CI: 1.1, 2.8]; *p* = 0.01) more likely to be observed in the 45–64-year age group and 240% more likely to be observed in the elderly (OR: 2.4 [95% CI: 1.5, 3.6]; *p* < 0.001), compared to young adults (18–44 years).

Detections of *C. minutissimum*, *Pseudomonas* or *S. aureus* were more frequently observed in males than females (Table 2); this difference was the most profound for *Pseudomonas* detections, where interdigital and plantar specimens from males were twice as likely to be detected with *Pseudomonas* than specimens from females (interdigital, OR: 2.2 [95% CI: 2.0, 2.4], *p* < 0.001; plantar, OR: 2.0 [95% CI: 1.6, 2.4], *p* < 0.001). Similar to the differential detection rates per age group observed for dermatophyte detections, *C. minutissimum* and *Pseudomonas* detections—from both interdigital and plantar foot specimens—were more commonly observed in young adults compared to other age groups. Specifically, interdigital and plantar foot specimens from children were 80% (OR: 0.2 [95% CI: 0.2, 0.4]; *p* < 0.001) and 70% (OR: 0.3 [95% CI: 0.1, 0.8]; *p* = 0.01), respectively, less likely to be detected to have *Pseudomonas* than specimens from young adults. The reverse trend was observed for *S. aureus*, where detection rates appeared to be inversely correlated with age. For interdigital *S. aureus* detections, specimens from children exhibited a 60% higher likelihood than young adults (OR: 1.6 [95% CI: 1.2, 2.1]; *p* = 0.002). For regional variations, the detection of *Pseudomonas* was more commonly observed in foot specimens submitted from the US Northeast (interdigital: 27.5%; plantar: 17.1%) and South (interdigital: 22.8%; plantar: 13.7%), compared to the US Midwest (interdigital: 18.4%; plantar: 9.6%). Specimens from the US Northeast also exhibited higher likelihoods for *C. minutissimum* detections.

### 3.3. Co-Infections of Bacteria and Fungus

Among tinea pedis patients with a confirmed detection of dermatophytes by PCR, the “dermatophytosis complex” variant (i.e., concomitant detection of bacteria) was found in an average of 45.5% (SD: 2.3) of specimens. Among these, the most common combination with dermatophytes was *S. aureus* (26.4% [SD: 1.5]), followed by *Pseudomonas* (18.5% [SD: 2.7]) and *C. minutissimum* (13.9% [SD: 2.4]). Combinations of dermatophytes with *C. minutissimum* or *Pseudomonas* were 30–40% more likely to be detected in interdigital specimens compared to plantar specimens, with males exhibiting a higher risk than females (Table 3). For concomitant detections of dermatophytes with *Pseudomonas*, children exhibited a reduced likelihood than young adults (OR: 0.3 [95% CI: 0.1, 0.8], *p* = 0.01), with the majority of specimens submitted from the US Northeast and South. For the combination of dermatophytes with *S. aureus*, no significant differences were found with regard to infection site and patient sex; the elderly exhibited a 50% lesser likelihood than young adults (OR: 0.5 [95% CI: 0.4, 0.6], *p* < 0.001) to be detected with this combination.

In the presence of *Candida*, a statistically significant higher detection rate of *Pseudomonas* was observed, compared to specimens without a co-detection of fungi or in dermatophyte-positive specimens (Figure 2). For interdigital foot specimens, the average detection rate for *Pseudomonas* was 37% (SD: 10.5) with a co-detection of *Candida*, compared to 23.4% (SD: 5.2) in fungi-negative specimens and 20.8% (SD: 4.5) in dermatophyte-positive specimens. For plantar foot specimens, the average detection rate of *Pseudomonas* was 27.4% (SD: 5.9) with a co-detection of *Candida*, compared to 12.1% (SD: 3.7) in fungi-negative specimens and 12.8% (SD: 3.6) in dermatophyte-positive specimens.

A co-detection of *Candida* and *Pseudomonas* was twice as likely to be observed in interdigital specimens compared to plantar specimens (OR: 2.0 [95% CI: 1.5, 2.7]; *p* < 0.001) (Table 4). Specimens from males exhibited a 60% higher likelihood than females overall (OR: 1.6 [95% CI: 1.3, 2.0]; *p* < 0.001). The US Northeast and South represented the highest proportion of specimens compared with the US Midwest and West.

### 3.4. Systematic Literature Review

Twenty-two studies, totaling 4968 interdigital or plantar foot specimens, reported the detection of fungal and/or bacterial etiological agents between 2010 and 2024 (Appendix A) [7,8,9,10,11,12,13,14,15,16,17,18,19,20,21,22,23,24,25,26,27,28]. The majority of the included studies reported findings from the general population (i.e., patients of all ages with and without comorbidities) seen at outpatient clinics. Three studies sampled asymptomatic individuals with clinically apparent healthy feet [14,23,26]. Two studies reported findings from the pediatric population [20,23]. Two studies compared findings from diabetic subjects against non-diabetic controls [24,25], and one study assessed findings from psoriasis and atopic dermatitis patients compared to control patients seen at a dermatology clinic [21].

Most studies relied on culture for pathogen identification; of these, four studies utilized PCR, sequencing or MALDI-TOF mass spectrometry to confirm the identity of the isolate [7,12,27,28]. One study reported the use of pyrosequencing to examine the skin microbiome in tinea pedis patients and healthy controls [14]. For dermatophytes, the reported prevalence of the *T. rubrum* complex, *T. mentagrophytes* complex and *Candida* was extracted and compared according to the region (Figure 3); countries were classified into the Global North and the Global South as per the United Nations. Studies conducted in the Global North demonstrated the predominance of the *T. rubrum* complex, while the *T. mentagrophytes* complex and *Candida* were rarely detected at <5%. In contrast, studies conducted in the Global South showed a statistically significant higher detection rate for the *T. mentagrophytes* complex (23.2% [95% CI: 9.0, 37.4] vs. 2.8% [95% CI: 0.3, 5.4]; *p* = 0.02). *Candida* spp. was also more frequently reported in the Global South, although this difference was not statistically significant.

## 4. Discussion

Tinea pedis is a common form of dermatophytosis with a growing morbidity burden affecting the elderly and in those with a history of diabetes, immunosuppression or peripheral vascular disease [2,4]. The introduction of multiplex qPCR diagnosis allows for an accurate and timely detection of the underlying fungal or bacterial pathogen, with the potential to reduce the risk of mistreatments in cases of bacterial colonization mimicking tinea pedis, and can help avoid treatment delays, as opposed to fungal cultures, which have an incubation period of 2–3 weeks.

The foot represents an area with diverse microbial niches attributed to differing moisture distributions and epidermis thickness [29]; for the toe web space, a thin epidermis, coupled with high moisture content and temperature, creates favorable growth conditions for bacteria and fungi. In this study, we observed a higher prevalence of *C. minutissimum* and *Pseudomonas* detected in the interdigital foot specimens compared to plantar foot specimens, reflecting two distinct microbial niches. Furthermore, we report a higher rate of *Pseudomonas* detection in the presence of *Candida*, which indicates potential species interactions. In the context of mucosal infections, a mixed infection of opportunistic pathogens—*C. albicans* and *P. aeruginosa*—has been associated with a worsened clinical prognosis due to the development of antifungal resistance, upregulation of virulence factors and exacerbation of the pro-inflammatory response [30].

In agreement with the literature, our results indicate that males have a higher risk than females for dermatophyte, *C. minutissimum*, *Pseudomonas* and *S. aureus* detections, which may be attributed to the use of occlusive footwear and hyperhidrosis [2]. More frequent hyperhidrosis in young adults may also help explain the higher detection rates observed in the 18–44-year age group for dermatophytes, *C. minutissimum* and *Pseudomonas*. For *Candida* detections, a higher prevalence was observed for elderly patients (≥65 years), which could be explained by a higher likelihood of skin trauma, obesity, diabetes and a weakened immune status [31]. The higher prevalence of *S. aureus* in children could be attributed to impetigo, which most frequently affects 2–5-year-olds [32]. However, due to low sample sizes, further investigation is needed to confirm this finding. Lastly, a significant regional variation was found in the US Northeast (*C. minutissimum* and *Pseudomonas*) and the US South (*Pseudomonas*). These variations may be linked to higher population densities or humidity levels, warranting further research.

A polymicrobial flora can often be found in the interdigital space, which can complicate the treatment of tinea pedis [4]. In this study, we found that close to half (45.5%) of the dermatophyte-positive specimens concurrently showed *C. minutissimum*, *Pseudomonas* and/or *S. aureus*, indicative of the “dermatophytosis complex” variant [4]. Mixed detections of dermatophytes with *C. minutissimum* or *Pseudomonas* were more commonly observed in interdigital foot specimens and in males. It is suspected that dermatophytes, through its keratinolytic activities, degrade the stratum corneum, thereby creating a point of entry for bacterial proliferation [5]; furthermore, certain dermatophyte strains have demonstrated the ability to produce penicillin-like antibiotics in vitro that increase the risk of resistant concomitant bacterial infections [33,34]. These findings are especially concerning for diabetic patients, where an interdigital foot infection can lead to leg erysipelas and hospitalizations due to severe ulcerations [5].

A review of the recent literature on the detection of fungal or bacterial agents in patients with suspected tinea pedis showed that most studies utilized the fungal culture method, which often precludes the detection of concomitant bacterial colonization due to the addition of antibiotics in the media. In a Turkish study, 17.4% (8/46) of interdigital dermatophyte isolates from patients presenting with foot lesions at a dermatology clinic were detected concurrently with *C. minutissimum* [19], which is similar to our finding (13.9%). In contrast, a Chinese study reported that *Corynebacterium* was only present in 3.3% (44/1324) of fungi-positive specimens obtained from suspected tinea pedis patients, while *S. epidermidis* (19.8% [262/1324]) and *S. aureus* (13.3% [176/1324]) were more commonly found as co-isolates [27]. In asymptomatic individuals, Sakka et al. raised the possibility of dermatophyte carriage (i.e., occult tinea pedis) where *T. rubrum* (12.2% [27/221])*, T. mentagrophytes* (0.9% [2/221]) and *E. floccosum* (0.9% [2/221]) were detectable in patients with the clinical appearance of normal feet [26]. However, conflicting data have been reported by Liu et al. using a metagenomic sequencing approach [14]. The results show that *T. rubrum* was only detected in tinea pedis patients (N = 26), while non-dermatophyte molds (*Phoma saxea*, *Aspergillus cibarius*) and yeasts (*C. halotolerans*, *Rhodotorula mucilaginosa*) were detected in healthy controls (N = 10). Another study also found no evidence of *T. rubrum* carriage in asymptomatic children (N = 865), although one *T. mentagrophytes* isolate was identified [23].

When examining interdigital foot specimens from diabetic patients, a higher prevalence of opportunistic pathogens such as *C. albicans*, *C. glabrata*, *C. tropicalis* and *C. krusei* has been reported compared to those obtained from non-diabetic patients [24]; bacterial colonization including *S. epidermidis* was also frequent [25]. Among dermatology outpatients, Leibovici et al. reported a higher prevalence of mycology-confirmed tinea pedis, mostly attributed to *T. rubrum*, in psoriasis patients compared to patients without concomitant dermatological diseases [21].

Although the *T. mentagrophytes* complex is considered an uncommon dermatophytic pathogen in the Western hemisphere, an infection by zoophilic *T. mentagrophytes* var. *mentagrophytes* can lead to acute inflammation [2,31]. According to studies conducted in Finland, Israel and Turkey, the *T. rubrum* complex was consistently reported as the dominant pathogen causing tinea pedis [19,20,21,26,28]. In contrast, mixed results were found in the Global South. Specifically, studies conducted in Egypt, Ethiopia, Iran and Senegal reported a significantly higher prevalence of the *T. mentagrophytes* complex [7,8,9,12,15,18,22]; in one study, *T. mentagrophytes, T. interdigitale* and *T. tonsurans* were isolated in half of the suspected tinea pedis patients (58.1% [36/62]) confirmed by ITS sequencing [12]. Studies conducted in China, India and Tunisia report the predominance of the *T. rubrum* complex [10,11,14,16,17,27], reflective of significant geographic variations affecting the circulation of dermatophytic pathogens.

In recent years, antifungal resistance in dermatophytoses has gained attention as an emerging public health concern, especially with regard to terbinafine. Case studies in Japan have identified resistant *T. rubrum* and *T. mentagrophytes* complex isolates from tinea pedis patients, with terbinafine MICs of 1 to >32 µg/mL [35,36]. In a US survey of podiatrists, half of the respondents reported that between 10% and >20% of tinea pedis patients experienced treatment failure, which was mostly attributed to terbinafine use (72.4%) [37]. Acquired resistance to terbinafine could be linked to subtherapeutic dosing with over-the-counter topical products, as well as empirical treatment practices without confirmatory testing. In a Danish survey, the majority of pharmacies (91%) recommended topical over-the-counter terbinafine to all patients suspected of tinea pedis without confirmatory testing, most pharmacies advised patients to only apply terbinafine to the affected foot area (82%), and they lacked awareness of antifungal resistance (88%) [38]. This concerning trend calls for continued advocacy and public education, as well as increased accessibility of diagnostic and resistance testing.

The findings presented in our study are limited by the convenience sampling approach and the retrospective data collection. The multiplex qPCR panel used in this study only allows for the detection of pan-dermatophytes and lacks genus- or species-level resolution. Patients’ medical histories were not available; hence, we cannot ascertain the generalizability of our findings to the outpatient population at large. As more advanced DNA-based diagnostic techniques become available, further research is warranted to update our understanding of common superficial fungal infections, especially given their rising global incidence and prevalence.

## 5. Conclusions

Tinea pedis is one of the most commonly encountered variants of dermatophytosis in clinical practice. Despite the introduction of PCR diagnosis enabling the rapid detection of fungal agents and concomitant bacterial colonization, there has been a paucity of studies examining the spectrum of pathogens found in tinea pedis patients using this methodology. In this study, we retrospectively examined 10,473 interdigital foot specimens and 3,956 plantar foot specimens submitted for multiplex qPCR testing in the US from 2020 to 2024. Our findings reaffirmed current knowledge on patient risk factors. Furthermore, the interdigital space showed a disproportionate proliferation of *C. minutissimum* and *Pseudomonas*, reflecting a distinct microbial niche. The detection rate of *Pseudomonas* was higher in the presence of *Candida*, indicating a higher risk of mixed infections. Among dermatophyte-positive specimens, a concomitant detection of bacteria (i.e., “dermatophytosis complex”)—including *S. aureus*, *Pseudomonas* and *C. minutissimum*—was found at a rate of 45.5%, which may be considered a negative prognostic factor that warrants the use of antibacterial agents especially concerning patients with diabetes. A literature review found significant geographical variations affecting the likelihood of finding the *T. mentagrophytes* complex in tinea pedis patients. In view of an increasingly globalized population, further prospectively designed surveillance studies are warranted to optimize the treatment of tinea pedis.

## Figures and Tables

**Figure 1 microorganisms-13-00184-f001:**
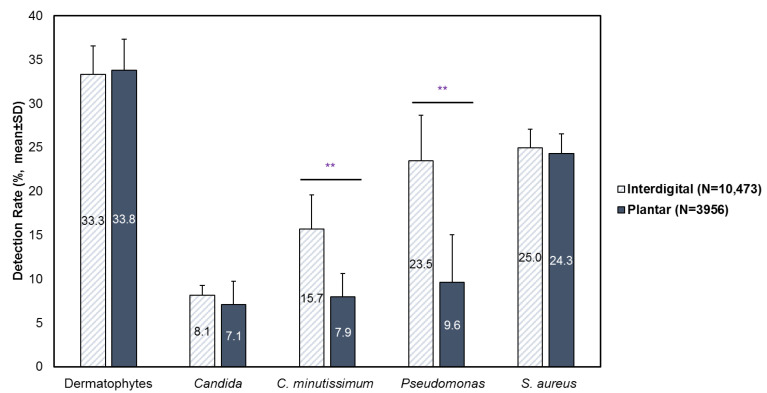
Prevalence of fungal and bacterial etiological agents in interdigital (N = 10,473) and plantar (N = 3956) foot specimens. Results are stratified by year and presented as mean ± SD. ** *p* < 0.01.

**Figure 2 microorganisms-13-00184-f002:**
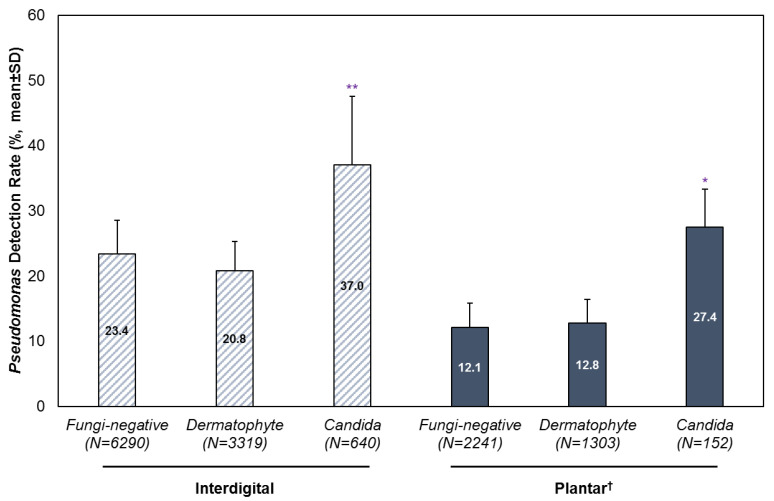
Detection of *Pseudomonas* with or without a co-detection of dermatophytes or *Candida*. Results are stratified by infection site and year, and presented as mean ± SD. ^†^ Analysis of plantar foot specimens excluded the years 2020 and 2021 due to low sample sizes. * *p* < 0.05; ** *p* < 0.01.

**Figure 3 microorganisms-13-00184-f003:**
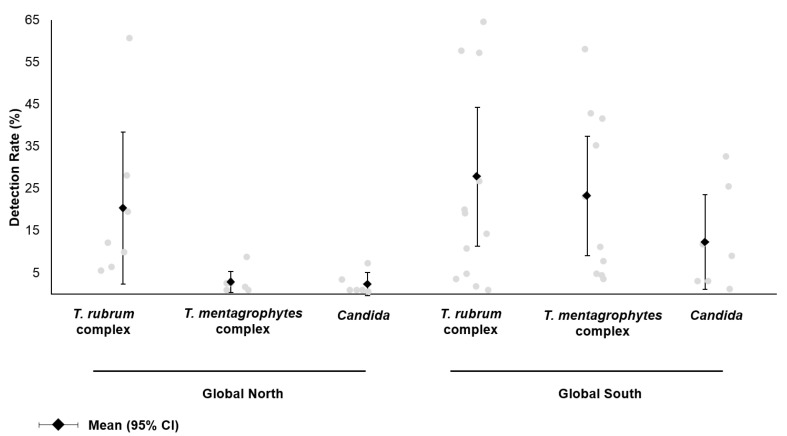
Reported prevalence of fungal etiological agents in patients with foot infections. From the literature review, studies reporting the identification of the *T. rubrum* complex (*T. rubrum*, *T. violaceum*), *T. mentagrophytes* complex (*T. mentagrophytes, T. interdigitale, T. tonsurans, T. benhamiae*) and *Candida* spp. were included. Results were stratified by the study location (Global North: Finland, Israel, Turkey; Global South: Botswana, China, Egypt, Ethiopia, India, Iran, Nepal, Senegal, Tunisia).

**Table 1 microorganisms-13-00184-t001:** Likelihoods of contracting an interdigital or plantar foot infection by dermatophytes or *Candida* stratified per patient sex, age group and location.

Parameter	Interdigital			Plantar		
	N	%	OR (95% CI)	N	%	OR (95% CI)
**Dermatophytes**						
**Sex**						
Male	2086	35.1	**1.1 (1.1, 1.2)**	777	39.6	**1.4 (1.2, 1.6)**
Female	1397	32.1	Referent	618	32.2	Referent
**Age Group, years**						
<18	64	29.0	**0.6 (0.4, 0.8)**	15	14.7	**0.3 (0.2, 0.5)**
18–44	947	40.5	Referent	347	38.9	Referent
45–64	1070	33.4	**0.7 (0.7, 0.8)**	428	36.0	**0.9 (0.7, 1.1)**
≥65	1460	31.1	**0.7 (0.6, 0.7)**	626	35.4	0.9 (0.7, 1.0)
**US Census Region**						
Northeast	801	34.1	0.9 (0.8, 1.1)	274	35.0	0.8 (0.6, 1.0)
Midwest	421	35.3	Referent	298	40.9	Referent
South	2060	33.4	0.9 (0.8, 1.0)	699	34.8	0.8 (0.6, 0.9)
West	257	33.8	0.9 (0.8, 1.1)	148	33.9	0.7 (0.6, 0.9)
** *Candida* **						
**Sex**						
Male	484	8.1	1.0 (0.8, 1.1)	120	6.1	1.2 (0.9, 1.6)
Female	365	8.4	Referent	100	5.2	Referent
**Age Group, years**						
<18	2	0.9	**0.1 (0.04, 0.6)**	NR	NR	NR
18–44	137	5.9	Referent	28	3.1	Referent
45–64	247	7.7	**1.3 (1.1, 1.7)**	65	5.5	**1.8 (1.1, 2.8)**
≥65	477	10.1	**1.8 (1.5, 2.2)**	125	7.1	**2.4 (1.5, 3.6)**
**US Census Region**						
Northeast	195	8.3	1.1 (0.9, 1.5)	50	6.4	1.3 (0.8, 2.0)
Midwest	88	7.4	Referent	37	5.1	Referent
South	530	8.6	1.2 (0.9, 1.5)	122	6.1	1.2 (0.8, 1.8)
West	51	6.7	0.9 (0.6, 1.3)	16	3.7	0.7 (0.4, 1.3)

ORs with statistically significant (*p* < 0.05) 95% CI are shown in bold. NR, not reported due to low sample size.

**Table 2 microorganisms-13-00184-t002:** Likelihoods of contacting an interdigital or plantar foot infection by *C. minutissimum*, *Pseudomonas* or *S. aureus* stratified per patient sex, age group and location.

Parameter	Interdigital			Plantar		
	N	%	OR (95% CI)	N	%	OR (95% CI)
** *C. minutissimum* **						
**Sex**						
Male	1039	17.5	**1.6 (1.4, 1.8)**	230	11.7	**1.7 (1.4, 2.2)**
Female	514	11.8	Referent	136	7.1	Referent
**Age Group, years**						
<18	40	18.1	1.1 (0.7, 1.5)	10	9.8	0.8 (0.4, 1.5)
18–44	402	17.2	Referent	110	12.3	Referent
45–64	508	15.8	0.9 (0.8, 1.0)	109	9.2	**0.7 (0.5, 1.0)**
≥65	625	13.3	0.7 (0.6, 0.8)	142	8.0	**0.6 (0.5, 0.8)**
**US Census Region**						
Northeast	439	18.7	**1.3 (1.0, 1.5)**	105	13.4	**1.9 (1.3, 2.6)**
Midwest	185	15.5	Referent	56	7.7	Referent
South	852	13.8	0.9 (0.7, 1.0)	180	9.0	1.2 (0.9, 1.6)
West	99	13.0	0.8 (0.6, 1.1)	31	7.1	0.9 (0.6, 1.4)
** *Pseudomonas* **						
**Sex**						
Male	1670	28.1	**2.2 (2.0, 2.4)**	322	16.4	**2.0 (1.6, 2.4)**
Female	654	15.0	Referent	171	8.9	Referent
**Age Group, years**						
<18	20	9.0	**0.2 (0.2, 0.4)**	5	4.9	**0.3 (0.1, 0.8)**
18–44	680	29.1	Referent	127	14.2	Referent
45–64	860	26.8	0.9 (0.8, 1.0)	180	15.1	1.1 (0.8, 1.4)
≥65	811	17.3	**0.5 (0.5, 0.6)**	198	11.2	**0.8 (0.6, 1.0)**
**US Census Region**						
Northeast	647	27.5	**1.7 (1.4, 2.0)**	134	17.1	**1.9 (1.4, 2.6)**
Midwest	219	18.4	Referent	70	9.6	Referent
South	1408	22.8	**1.3 (1.1, 1.5)**	275	13.7	**1.5 (1.1, 2.0)**
West	94	12.4	**0.6 (0.5, 0.8)**	32	7.3	0.7 (0.5, 1.1)
** *S. aureus* **						
**Sex**						
Male	1601	26.9	**1.2 (1.1, 1.4)**	519	26.4	**1.3 (1.1, 1.5)**
Female	995	22.8	Referent	421	21.9	Referent
**Age Group, years**						
<18	97	43.9	**1.6 (1.2, 2.1)**	NR	NR	NR
18–44	778	33.3	Referent	274	30.7	Referent
45–64	845	26.3	**0.7 (0.6, 0.8)**	288	24.2	**0.7 (0.6, 0.9)**
≥65	915	19.5	**0.5 (0.4, 0.5)**	353	20.0	**0.6 (0.5, 0.7)**
**US Census Region**						
Northeast	601	25.6	1.0 (0.8, 1.1)	190	24.3	1.1 (0.8, 1.3)
Midwest	313	26.2	Referent	170	23.4	Referent
South	1506	24.4	0.9 (0.8, 1.0)	494	24.6	1.1 (0.9, 1.3)
West	215	28.3	1.1 (0.9, 1.4)	99	22.7	1.0 (0.7, 1.3)

ORs with statistically significant (*p* < 0.05) 95% CI are shown in bold. NR, not reported due to low sample size.

**Table 3 microorganisms-13-00184-t003:** Likelihoods of detecting dermatophytes with concomitant bacterial colorizations (dermatophytosis complex).

Parameter	Mixed Detection		
	N	%	OR (95% CI)
**Dermatophytes and *C. minutissimum***			
**Sex**			
Male	228	2.9	**1.4 (1.1, 1.7)**
Female	134	2.1	Referent
**Age Group, years**			
<18	5	1.5	0.6 (0.2, 1.4)
18–44	87	2.7	Referent
45–64	108	2.5	0.9 (0.7, 1.2)
≥65	166	2.6	1.0 (0.7, 1.2)
**US Census Region**			
Northeast	86	2.7	1.0 (0.7, 1.5)
Midwest	51	2.7	Referent
South	206	2.5	0.9 (0.7, 1.3)
West	24	2.0	0.8 (0.5, 1.2)
**Infection Site**			
Interdigital	286	2.7	**1.3 (1.0, 1.7)**
Plantar	81	2.0	Referent
**Dermatophytes and *Pseudomonas***			
**Sex**			
Male	331	4.2	**1.8 (1.4, 2.1)**
Female	152	2.4	Referent
**Age Group, years**			
<18	4	1.2	**0.3 (0.1, 0.8)**
18–44	138	4.3	Referent
45–64	163	3.7	0.9 (0.7, 1.1)
≥65	186	2.9	**0.7 (0.5, 0.8)**
**US Census Region**			
Northeast	137	4.4	**1.8 (1.3, 2.6)**
Midwest	47	2.4	Referent
South	293	3.6	**1.5 (1.1, 2.0)**
West	13	1.2	**0.5 (0.3, 0.9)**
**Infection Site**			
Interdigital	387	3.7	**1.4 (1.1, 1.8)**
Plantar	105	2.7	Referent
**Dermatophytes and *S. aureus***			
**Sex**			
Male	442	5.6	1.1 (0.9, 1.2)
Female	327	5.2	Referent
**Age Group, years**			
<18	20	6.2	0.8 (0.5, 1.2)
18–44	260	8.0	Referent
45–64	245	5.6	**0.7 (0.6, 0.8)**
≥65	261	4.0	**0.5 (0.4, 0.6)**
**US Census Region**			
Northeast	183	5.8	0.9 (0.7, 1.1)
Midwest	123	6.4	Referent
South	415	5.1	**0.8 (0.6, 1.0)**
West	64	5.3	0.8 (0.6, 1.1)
**Infection Site**			
Interdigital	590	5.6	1.1 (1.0, 1.4)
Plantar	196	5.0	Referent

ORs with statistically significant (*p* < 0.05) 95% CI are shown in bold.

**Table 4 microorganisms-13-00184-t004:** Likelihoods of a mixed detection of *Candida* and *Pseudomonas* stratified per patient sex, age group, location and sampling site.

Parameter	*Candida* and *Pseudomonas*		
	N	%	OR (95% CI)
**Sex**			
Male	220	2.8	**1.6 (1.3, 2.0)**
Female	109	1.7	Referent
**Age Group, years**			
<18	0	0	-
18–44	72	2.2	Referent
45–64	117	2.7	1.2 (0.9, 1.6)
≥65	151	2.3	1.0 (0.8, 1.4)
**US Census Region**			
Northeast	103	3.3	**2.1 (1.4, 3.2)**
Midwest	30	1.6	Referent
South	195	2.4	**1.5 (1.0, 2.3)**
West	12	1.0	0.6 (0.3, 1.3)
**Infection Site**			
Interdigital	285	2.7	**2.0 (1.5, 2.7)**
Plantar	55	1.4	Referent

ORs with statistically significant (*p* < 0.05) 95% CIs are shown in bold.

## Data Availability

Restrictions apply to the availability of these data. Data were obtained from Bako Diagnostics (Alpharetta, GA, USA) and are available from A.K.G with the permission of Bako Diagnostics.

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
