# Peer review of "Interdigital and Plantar Foot Infections: A Retrospective Analysis of Molecularly Diagnosed Specimens in the United States and a Literature Review"

_microorganisms, 2025, doi:10.3390/microorganisms13010184_

Round 1
Reviewer 1 Report
Comments and Suggestions for Authors
This is an elegant manuscript concerning a retrospective analysis of interdigital and plantar foot infections in the USA.
Tinea pedis (athlete's foot) affects up to 25% of the U.S. population, with interdigital tinea pedis being the most common form. This condition can cause skin damage, chronic lesions, and significant morbidity. A "dermatophytosis complex," involving fungal infections with concurrent bacterial colonization, complicates treatment. In this study, the authors analyzed 14,429 foot skin specimens using multiplex qPCR. The study emphasizes the value of PCR diagnostics in detecting fungal and bacterial pathogens for timely and precise treatment.
The references are updated and the study is well-organized.
The methodology is well organized and described. The results support the discussion and the conclusion. In my opinion, there are two or three minor improvements to be made:
1) In my opinion the title should be “Interdigital and plantar foot infections: A retrospective analysis of molecularly diagnosed specimens in the United States and literature review” or even better (please see comment 3) “Interdigital and plantar foot infections: literature review and retrospective analysis of molecularly diagnosed specimens in the United States”
2) This comment is a personal opinion that the authors should think but they are, of course, free to decide what they think is better. If I were the authors, I would prefer to place the literature review in a results section right at the beginning, titled Systematic Literature Review. Only afterward would I include the study.
3) In the results section I would suggest different subtitles. Depending on following or not comment 2.
According to comment 2:
3.1. Systematic literature review (lines 221-254)
3.2. Microorganisms (lines 128-140)
3.3. Patients influence (lines 142-182)
3.4. Co-infections of bacteria and fungus (lines 183-221)
Not following comment 2:
3.1. Microorganisms (lines 128-140)
3.2. Patients influence (lines 142-182)
3.3. Co-infections of bacteria and fungus (lines 183-221)
This is an elegant work that after these changes is publishable.
Author Response
Reviewer 1 comments
This is an elegant manuscript concerning a retrospective analysis of interdigital and plantar foot infections in the USA.
Authors: Thank you for taking the time to review our work.
Tinea pedis (athlete's foot) affects up to 25% of the U.S. population, with interdigital tinea pedis being the most common form. This condition can cause skin damage, chronic lesions, and significant morbidity. A "dermatophytosis complex," involving fungal infections with concurrent bacterial colonization, complicates treatment. In this study, the authors analyzed 14,429 foot skin specimens using multiplex qPCR. The study emphasizes the value of PCR diagnostics in detecting fungal and bacterial pathogens for timely and precise treatment.
The references are updated and the study is well-organized.
Authors: We appreciate your appraisal of our work.
The methodology is well organized and described. The results support the discussion and the conclusion. In my opinion, there are two or three minor improvements to be made:
1) In my opinion the title should be “Interdigital and plantar foot infections: A retrospective analysis of molecularly diagnosed specimens in the United States and literature review” or even better (please see comment 3) “Interdigital and plantar foot infections: literature review and retrospective analysis of molecularly diagnosed specimens in the United States”
Authors: Thank you for the comment. We have revised the manuscript title as follows: “Interdigital and plantar foot infections: A retrospective analysis of molecularly diagnosed specimens in the United States and literature review”.
2) This comment is a personal opinion that the authors should think but they are, of course, free to decide what they think is better. If I were the authors, I would prefer to place the literature review in a results section right at the beginning, titled Systematic Literature Review. Only afterward would I include the study.
Authors: Thank you for your suggestion. When we planned on conducting a systematic literature review, it was intended to supplement the findings from our original data and to provide context for the results. Therefore, we would prefer to place the literature review sub-section toward the end of the results section.
3) In the results section I would suggest different subtitles. Depending on following or not comment 2.
According to comment 2:
3.1. Systematic literature review (lines 221-254)
3.2. Microorganisms (lines 128-140)
3.3. Patients influence (lines 142-182)
3.4. Co-infections of bacteria and fungus (lines 183-221)
Not following comment 2:
3.1. Microorganisms (lines 128-140)
3.2. Patients influence (lines 142-182)
3.3. Co-infections of bacteria and fungus (lines 183-221)
Authors: We have re-organized the sub-headings as suggested: 3.1. Microorganisms; 3.2: Patients’ influence; 3.3: Co-infections of bacteria and fungus; 3.4: Systematic literature review
This is an elegant work that after these changes is publishable.
Authors: Thank you again for taking the time to look over our manuscript. We hope that the changes made were acceptable.
Reviewer 2 Report
Comments and Suggestions for Authors
1. abstract needs to be shorter to make it more appealing for future readers
2. aim does not provide information about literature review you conducted in methods
3. methods could be improved to make it more clear what exactly you did
4. table 5 can be supplementary material
5. I find it unusual to have 34 references only, please try to include more recently published data
6. other figures and tables are nicely presented and I do find this study to be relevant in the field. I also believe it will add to the body of literature on a matter of tinea pedis
Author Response
Reviewer 2 comments
- abstract needs to be shorter to make it more appealing for future readers
Authors: We agree that there should be more brevity in the abstract. We have reduced the word count from 348 to 288.
- aim does not provide information about literature review you conducted in methods
Authors: Thank you for pointing out our oversight. We have revised the Introduction section accordingly (lines 80-82).
- methods could be improved to make it more clear what exactly you did
Authors: Thank you for your comment. For proprietary reasons, we are unable to provide additional information with regards to the DNA extraction and the multiplex qPCR assay used in this study. All assays were performed at a commercial laboratory with CLIA (Clinical Laboratory Improvements Amendments) certification (Bako Diagnostics, GA, USA) in compliance with federal regulations.
- table 5 can be supplementary material
Authors: Thank you for the suggestion. We have moved Table 5 to the Supplementary Materials section (Lines 380-382)
- I find it unusual to have 34 references only, please try to include more recently published data
Authors: Thank you for your comment. In this study, a systematic literature search was conducted in October 2024 to review all recent clinical investigations (2010-2024) of tinea pedis among the general population. After applying our exclusion criteria, a total of 22 studies were identified and summarized. Since our manuscript is not intended as a standalone review article, we used more stringent selection criteria to supplement the findings from our original data. We agree with reviewer that more recently published data could be included to better inform the reader. With regards to antifungal resistance, which is an emerging public health concern, we have added a paragraph in the Discussion section (Lines 338-351).
- other figures and tables are nicely presented and I do find this study to be relevant in the field. I also believe it will add to the body of literature on a matter of tinea pedis
Authors: Thank you for taking the time to review our work.